# CD26 Deficiency Controls Macrophage Polarization Markers and Signal Transducers during Colitis Development and Resolution

**DOI:** 10.3390/ijms23105506

**Published:** 2022-05-14

**Authors:** Iva Vukelic, Suncica Buljevic, Lara Baticic, Karmela Barisic, Barbara Franovic, Dijana Detel

**Affiliations:** 1Department of Medical Chemistry, Biochemistry and Clinical Chemistry, Faculty of Medicine, University of Rijeka, Brace Branchetta 20, 51000 Rijeka, Croatia; iva.vukelic@uniri.hr (I.V.); suncica.buljevic@uniri.hr (S.B.); lara.baticic@uniri.hr (L.B.); kbarisic@pharma.hr (B.F.); 2Department of Medical Biochemistry and Hematology, Faculty of Pharmacy and Biochemistry, University of Zagreb, Ante Kovacica 1, 10000 Zagreb, Croatia; barbara.franovic@student.uniri.hr

**Keywords:** ulcerative colitis, dipeptidyl peptidase 4, CD26 deficiency, macrophage polarization, STAT molecules

## Abstract

Ulcerative colitis (UC) is a multifactorial condition characterized by a destructive immune response that failed to be attenuated by common regulatory mechanisms which reduce inflammation and promote mucosa healing. The inhibition of CD26, a multifunctional glycoprotein that controls the immune response via its dipeptidyl peptidase (DP) 4 enzyme activity, was proven to have beneficial effects in various autoimmune inflammatory diseases. The polarization of macrophages into either pro-inflammatory M1 or anti-inflammatory M2 subclass is a key intersection that mediates the immune-inflammatory process in UC. Hence, we hypothesized that the deficiency of CD26 affects that process in the dextran sulfate sodium (DSS)-induced model of UC. We found that mRNA expression of M2 markers arginase 1 and Fizz were increased, while the expression of M1 marker inducible NO synthase was downregulated in CD26^−/−^ mice. Decreased STAT1 mRNA, as well as upregulated pSTAT6 and pSTAT3, additionally support the demonstrated activation of M2 macrophages under CD26 deficiency. Finally, we investigated DP8 and DP9, proteins with DP4-like activity, and found that CD26 deficiency is not a key factor for the noted upregulation of their expression in UC. In conclusion, we demonstrate that CD26 deficiency regulates macrophage polarization toward the anti-inflammatory M2 phenotype, which is driven by STAT6/STAT3 signaling pathways. This process is additionally enhanced by the reduction of M1 differentiation via the suppression of proinflammatory STAT1. Therefore, further studies should investigate the clinical potential of CD26 inhibitors in the treatment of UC.

## 1. Introduction

Inflammatory bowel disease (IBD) is a multifactorial disease characterized by chronic inflammation of the gastrointestinal tract and includes ulcerative colitis (UC) and Crohn’s disease (CD). A common feature of both entities is the initial destruction of intestinal epithelium followed by an excessive and dysregulated activation of the immune system [1]. The production of numerous effector molecules leads to the development of an uncontrolled inflammatory response. Despite intensive research and significant advances in understanding the mechanisms of chronic inflammation, the exact etiology and pathogenesis of UC remain unclear.

In recent years, macrophages have been identified as essential elements that not only maintain intestinal homeostasis but also the inflammatory response and mucosal healing, key steps in completing remission [2]. Depending on the microenvironment, macrophages are polarized into one of the two functionally distinct forms: classically activated (M1) macrophages, which contribute to the pathogenesis of the disease by secreting proinflammatory cytokines and causing tissue damage, and alternatively activated (M2) macrophages, which secrete anti-inflammatory factors involved in resolving inflammation [3]. As the imbalance of intestinal macrophage populations could modify the course of inflammation, the regulation of M1/M2 balance has recently been targeted as a potential therapeutic strategy for UC [4]. M1 activated macrophages are characterized by an increase in the expression of proinflammatory cytokines such as tumor necrosis factor-alpha (TNFα), interleukin (IL)-6, and inducible NO synthase (iNOS), which are considered typical M1 markers [5]. A polarization toward M2 macrophages is characterized by an increase in several markers, including arginase 1 (Arg-1) and Fizz [6], which were found to be increased in experimental colitis [7]. It was shown that increased Arg-1 activity results in the production of polyamines and collagen and favors tissue remodeling and wound healing. The role of intestinal macrophages and their contribution to the development of intestinal inflammation is regulated by several factors, which, among others, include chemoattractants such as monocyte chemoattractant protein (MCP) 1 and macrophage inflammatory protein-1 alpha (MIP1α) [8]. Interest in the involvement of MCP-1 in UC pathology arose after increased expression of MCP-1 in the colonic mucosa was observed in IBD patients [9]. MIP1α is a chemoattractant suggested to play an important role in the inflammatory process in general by promoting the recruitment of leukocytes to the site of inflammation [10] and, like MCP-1, it was found to be increased in patients with IBD [9].

M1/M2 polarizing pathways are based on the balance between members of the signal transducer and activator of transcription (STAT) family of proteins, mainly STAT1, STAT3, and STAT6, and IRF/STAT signaling is the central pathway that controls M1/M2 polarization [3]. IFN-γ-dependent STAT1 activation has been linked to M1 polarization, while IL-4 and IL-13 stimulated activation of STAT6 is associated with M2 polarization, although both M1 and M2 can also be activated by IL-10 induced activation of STAT3 [3,8]. In addition, STAT1 has been shown to inhibit STAT6 activation, thus attenuating the anti-inflammatory action of M2 macrophages. Furthermore, all three mentioned STATs are involved in the pathogenesis of IBD with the final effect that can be either pro- or anti-inflammatory, depending on their function in different cell types of adaptive and innate immunity [11,12].

CD26 is an immune cell surface protein with dipeptidyl peptidase (DP) 4-like enzyme activity that participates in glucose homeostasis and is already used in the therapy of diabetes mellitus [13]. However, the role of this protein has been limitedly described in the modulation of the immune response in numerous diseases, especially autoimmune and inflammatory, which are shown to be associated with an alteration of DP4 expression and/or activity [13,14,15]. In this regard, several studies have shown that inhibition of DP4 is beneficial in the treatment of IBD, including experimental colitis [15]. Lately, there are indications of the involvement of DP8 and DP9, the other two members of the DP protein family, in the regulation of immune-inflammatory response, especially as changes in their expressions have been observed in a broad spectrum of diseases: both enzymes are upregulated in asthma [16], and the expression of DP8 is 8-fold increased in the colon of mice with chemically induced colitis [17]. Furthermore, it was shown that the abolishment of DP9 activity or DP8/9 inhibition attenuated proinflammatory activation of human M1 macrophages [18] and decreased the secretion of M1 proinflammatory cytokines [19]. Our previous research showed that experimental IBD in mice causes alterations in DP8 and DP9 mRNA expression, concentration, and enzymatic activity at both systemic and tissue levels [20] but also increases the number of macrophages in colonic tissue during the acute phase of dextran sulfate sodium (DSS)-induced colitis [21]. Therefore, it can be assumed that through the modulation of activation and polarization of macrophages, DP8 and DP9 could be involved in the dysregulated immune response in UC.

As macrophage functions range from pro- to anti-inflammatory, this study aimed to analyze a group of selected chemoattractants, M1/M2 macrophage polarization markers, and transcriptional factors in a chemically induced UC model to elucidate the impact of CD26 deficiency on their expression patterns. Additionally, our goal was to assess changes in the expression pattern of DP8 and DP9 in the colonic tissue during the development and resolution of experimental colitis under CD26 deficiency.

## 2. Results

### 2.1. CD26 Deficiency Attenuates Clinical Symptoms of Colitis and Improves Histological Damage of Colonic Tissue

C57BL/6 and CD26^−/−^ mice treated with 3% DSS solution for 7 days effectively developed colitis determined by observation of clinical parameters and calculation of DAI, as well as pathohistological analysis. During the development of colitis, both mouse strains exhibited a decrease in body weight (Figure 1b) which was most pronounced on day 7.

The loss of body weight in CD26^−/−^ mice was less prominent, but interestingly, the recovery to initial values was prompter compared to C57BL/6 mice. Rectal bleeding score (Figure 1c), diarrhea score (Figure 1d), and DAI score (Figure 1e) were also increased in both strains on day 7 of the experiment without any significant difference between strains. Control mice that received tap water did not develop previously mentioned clinical symptoms. Colon length decreased as the disease progressed and colon shortening was significantly accentuated in both C57BL/6 and CD26^−/−^ mice on days 3 and 7 (Figure 1f,g). However, CD26 deficiency significantly mitigated colon shortening compared to C57BL/6 at the peak of inflammation on day 7. Histological examination of colonic tissue samples shown in Figure 1i revealed the appearance of pathological changes starting on day 3. Decreases in crypts’ width, focal areas of mild edema, and inflammatory cell infiltration into the connective tissue of lamina propria and submucosa were observed. With the aggravation of inflammation, further destruction of mucosal architecture with marked colonic damage on day 7 was noticed, changes being slightly less present in CD26^−/−^ mice as evidenced by a decrease in microscopic damage score (Figure 1h). Histopathological changes were accompanied by a progressive increase in inflammatory cell infiltration, distortion of crypt architecture, depletion of goblet cells, and extensive submucosal edema. On day 15, when clinical signs of inflammation disappeared completely, the severity of inflammatory lesions decreased in both mouse strains. Restricted crypt erosion sites showed reepithelization and formation of new crypts. In line with the histological changes, immunohistochemical analysis of Ki-67 revealed a significant decrease in the number of Ki-67 positive crypt cells in the acute phase of colitis (day 7) in both mouse strains relative to their respective controls (Figure 2b). However, no changes were observed between CD26^−/−^ and C57BL/6 mice. In the normal colonic mucosa, Ki-67 positive cells were mainly distributed in the basal zone of the intestinal crypts and rarely in the surface epithelium. On the other hand, in the acute phase of colitis (day 7), the Ki-67 expression was mainly associated with inflammatory cells (Figure 2a).

### 2.2. CD26 Deficiency Has Diverse Effects on the Expression of Macrophage Chemoattractants MCP-1 and MIP1α and Proinflammatory TNFα Cytokine

Macrophage migration into the inflamed colon caused by DSS is likely to be mediated by chemoattractants such as MCP-1 and MIP1α; therefore, we determined their concentration and mRNA expression. As shown in Figure 3a, DSS-induced colitis caused an increase in MCP-1 concentration in the colonic tissue of both strains during inflammation and the process of healing (day 15). However, the increase was more pronounced in CD26^−/−^ mice and more than doubled on day 7. Circulating MCP-1 levels were increased at the onset and peak of inflammation, but it was observed that CD26 deficiency did not cause any change in this pattern of MCP-1 serum concentrations (Figure 3b). The expression of MCP-1 and MIP1α mRNA in colonic tissue had a very similar pattern of change. Both genes were upregulated in both strains during the whole experimental period, most prominently on day 7. This increase was more accentuated in CD26^−/−^ mice at the peak of inflammation when it was almost two times higher compared to C57BL/6 mice (Figure 3c,d). TNFα is a classical proinflammatory cytokine that mediates intestinal inflammation, and its expression increases in IBD. It was shown that CD26 inhibition might abolish or stimulate TNFα production [22]. Therefore, we wanted to examine whether CD26 deficiency affects its gene expression. TNFα mRNA level increased with the inflammation onset and reached its peak on day 7, with values returning to normal during the resolution of colitis in both strains (Figure 3e). In CD26^−/−^ mice, however, TNFα increase was only 50% of the value measured in C57BL/6 mice at both time points representing the inflammatory phase, thus suggesting a weaker involvement of this particular cytokine in the inflammatory response under CD26 deficiency.

### 2.3. CD26 Deficiency Negatively Regulates iNOS Expression but Increases the Expression of M2 Macrophage Markers Arg-1 and Fizz

The expression of iNOS, a representative marker of M1 macrophages, increased almost 30 times in C57BL/6 mice during acute inflammation, but no such upregulation was noted in CD26^−/−^ mice (Figure 4a). On the other hand, an increase in mRNA level of M2 marker Arg-1 was present in both strains during the whole experimental period in comparison to the control group, as can be seen in Figure 4b. However, this increase was significantly greater in CD26^−/−^ than in C57BL/6 mice at the peak of inflammation, as well as during the recovery period. The expression of intestinal Fizz mRNA changed far less pronouncedly. DSS-induced colitis caused an increase in Fizz expression on day 3 in C57BL/6 mice, but, interestingly, no increase was observed in acute inflammation on day 7 (Figure 4c), while in CD26-deficient mice, Fizz expression increased during the whole period of acute inflammation.

### 2.4. CD26 Deficiency Modulates STA6, STAT3, and STAT1 Gene and Protein Expression in Colitis

Our results revealed an interesting pattern of relative changes in STAT1 mRNA expression in colonic tissue of analyzed mouse strains (Figure 5a). While C57BL/6 mice showed an increase in mRNA expression on days 3 and 7, in CD26^−/−^ mice, such changes were not observed, and STAT1 mRNA expression remained constant during the whole experimental period. However, protein expressions of phospho-STAT1 were increased on both days 3 and 7 in CD26^−/−^ mice (Figure 5b). On the other hand, STAT3 mRNA expression changed equally in both C57BL/6 and CD26^−/−^ mice, showing a significant increase during the onset and development of colitis (days 3 and 7), with its peak on day 7 and the attenuation of the increase on day 15 (Figure 5a). A similar pattern was also observed at the protein level of STAT3 and phospho-STAT3 with a slightly more intense increase in phospho-STAT3 under CD26 deficiency on both days 3 and 7 (Figure 5b). Intriguingly, an increase in STAT6 gene expression was seen on day 3 but not on day 7 in C57BL/6 mice, unlike the two previously mentioned transcription factors. CD26 deficiency affects this signaling pathway as the expression of colonic STAT6 was increased on day 7, as well as on day 3 in CD26^−/−^ mice. Phospho-STAT6 protein expression is under the influence of CD26 deficiency as significant differences in all time points were present, with an emphasis on day 7 (Figure 5b). As shown in Figure 5c, the ratio between phosphorylated and total STATs’ protein expressions further supports that finding (Figure 5c).

### 2.5. DP8 and DP9 Are Upregulated in DSS-Induced Colitis Yet Unaffected by CD26 Deficiency

The analysis of the impact of CD26 deficiency on the expression patterns of DP8 and DP9 is shown in Figure 6. DP8 mRNA expression increased in both mouse strains upon colitis development, slightly more pronounced in CD26^−/−^ mice (Figure 6a). On day 7 of the experimental period, DP8 expression in C57BL/6 mice was almost similar to the control group, but in CD26^−/−^ mice, the increase seen on day 3 was still present. Changes in DP9 gene expression were generally in line with these results (Figure 6b).

DP9 mRNA level was increased in both mouse strains on days 3 and 7 with no significant differences between the strains. To obtain additional information regarding the expression and localization of DP8 and DP9, we performed an immunohistochemical analysis of colonic tissue (Figure 6c,d). In line with the results of mRNA expression, the amount of DP8 and DP9 positively stained cells were visibly greater in colonic tissue of C57BL/6 and CD26^−/−^ on day 7 of the experimental period than in respective control mice and both proteins were markedly expressed in colonic epithelial cells.

## 3. Discussion

UC is a chronic progressive disease characterized by the development of an exaggerated and aberrant inflammatory response that results in tissue damage [1]. The excessive immune-inflammatory response is driven by multiple interactions between adaptive and innate immune cells, their soluble mediators, and intestinal structural components present in the inflammatory microenvironment. Elucidation of the mechanisms of the immune-inflammatory pathways underlying the pathogenesis of UC has developed rapidly over the last decade, intending to develop new and more effective therapies.

CD26 is a ubiquitous protein that acts as a receptor with a costimulatory function and an enzyme that possesses DP4 cleavage properties. It is abundantly expressed on epithelial and endothelial cells of the colon but also on immune cells [13]. Previous studies have shown attenuation of pathophysiological processes by DP4 inhibition in various diseases [23], including UC [24], indicating a protective role of DP4 inhibition. The results of our study confirmed this beneficial effect of CD26 deficiency, demonstrating that the lack of CD26 improved the overall clinical picture by promoting faster regaining of mice body weight and decreasing DAI. Likewise, the histopathological analysis showed less pronounced damage to colonic tissue in CD26^−/−^ mice compared to C57BL/6 mice in the acute phase of colitis, evidenced by a lower microscopic damage score. However, it should be emphasized that the observed effects were relatively slight for certain time points and were not noted for all parameters at the same level of significance. Structurally related to DP4 and with overlapping proteolytic activity are DP8 and DP9 proteins, which, due to their different localization in the cell and expression compared to DP4, may exert different functions [25]. Abundant DP8 and DP9 have been found in the colon and their functions described so far show that they are involved in cell-extracellular matrix interactions, proliferation, and apoptosis, as well as in immunoregulation and inflammatory response [12,26]. Regarding their role in the inflammatory response, the involvement of DP8, but not DP9, in colitis has been demonstrated [17]. According to our results, DP8 expression was significantly increased in CD26^−/−^ mice in acute colitis in comparison to the untreated group but not between C57BL/6 and CD26^−/−^. This was not observed during the period of healing, whereas DP9 expression was increased in both strains with no differences among them. Two conclusions can be drawn from such expression profiles. First, DP8 upregulation could be a part of the host’s compensatory reaction to the deficiency of CD26 in the acute inflammatory response. Second, it can be assumed that DP9 alone, rather than DP8, participates in the regulation of UC inflammation in the context of the proposed role of DP9 in the regulation of macrophage activation as DP9 was detected in macrophage-rich regions [18]. Upregulated DP9 was shown in both proinflammatory M1 and anti-inflammatory M2 macrophages after monocyte to macrophage differentiation [18] and, according to confirmed increased number of macrophages in the colonic mucosa of CD26^−/−^ mice in the acute phase of colitis in our previous research [21], it can be suggested that DP9 is involved in the regulation of macrophage activation in colitis. Moreover, inactivation of M1 macrophages after DP9 inhibition has been shown [18], so further studies of macrophage polarization toward M2 phenotype under the synergistic effect of CD26 deficiency and DP9 inhibition in UC treatment are of great importance with the assumption that this double inhibition may accelerate the healing process.

High upregulation of MCP-1 and MIP1α observed in CD26^−/−^ animals on day 7, when the inflammation was at its highest level, suggested that CD26 deficiency enhances macrophage migration to the inflammation site during acute colitis, thus supporting our previous research [21], but also suggesting a correlation between CD26, MCP-1, and MIP1α. Previous research has shown that MCP-1 acts as a chemoattractant for monocytes and some other immune cells, such as memory T lymphocytes and natural killer cells, and mediates the recruitment of monocytes and macrophages from the bloodstream to the inflamed tissue, which is one of the crucial events in the pathogenesis of a variety of immune-mediated diseases, including IBD [27]. Furthermore, the binding of MCP-1 to C-C chemokine receptor 2 (CCR2) expressed on monocytes, neutrophils, and lymphocytes have been reported to promote Th1 immune responses in vivo [27]. Accordingly, it has been shown that CCR2-deficient mice were protected from DSS-induced intestinal adhesions and mucosal ulcerations [27]. It was shown that the increase in serum concentration of MIP1α exacerbates immune-mediated colitis [28]. In addition to the chemokines, the immune response represents different actions of multiple cytokines, including a classical proinflammatory cytokine TNFα, a hallmark of inflammation, significantly implicated in the pathogenesis of IBD. Interestingly, despite the theory that proinflammatory cytokines are the main inducers of MCP-1, our analysis of TNFα expression during colitis development and resolution showed almost a 2-fold reduced TNFα mRNA level in CD26^−/−^ mice compared to C57BL/6 animals in the acute phase of colitis. The same effect was previously noted in vitro when an addition of the selective DP4 inhibitor sitagliptin inhibited the increase in TNFα mRNA and protein expression in isolated human monocytes [29], suggesting a connection between the lack of functional CD26 and suppression of TNFα effects. As TNFα upregulates other proinflammatory mediators such as IL-6 and IL-1β, consequently enhancing the early sequences of the inflammatory cascade [30], the decrease in TNFα gene expression in the absence of CD26 obtained by our results may favor a reduction in the inflammatory response. This could be one of the reasons for the observed decrease in disease activity and improvement of the histological picture of the colon in CD26^−/−^ mice.

Previous research has shown an association between increased iNOS activation and the production of NO during the initiation and progression of inflammation in the gastrointestinal tract, which represents a segment of the rapid intestinal antibacterial response [31]. There is accumulating evidence that the iNOS gene is overexpressed in various forms of mucosal inflammation, including DSS-induced colitis [32], also confirmed by our result of highly upregulated iNOS in C57BL/6 but not in CD26^−/−^ mice. The progression of inflammation was also characterized by an increase in the expression of the Arg-1 gene in both mouse strains but more pronounced in CD26^−/−^ mice. According to a widely accepted model, Arg-1 reduces the NO synthesis by consuming iNOS’s substrate L-arginine [33]. As murine M1 macrophages produce high levels of NO by overexpressing iNOS, and activation of M2 anti-inflammatory macrophages is characterized by an increased expression of Arg-1 [3], the concept by which the competition of Arg-1 and iNOS for L-arginine would regulate the macrophage balance between M1 and M2 phenotypes are proposed. Following that, our results of decreased expression of iNOS and increased expression of Arg-1 under CD26 deficiency indicated activation of M2 macrophages and consequential attenuation of inflammation [33]. Furthermore, iNOS production by M1 macrophages is mediated by cytokines such as IFN-γ and TNFα [3], while the transcriptional activation of the Arg-1 gene is mediated by activated STAT6 [34]. Therefore, our result can be interpreted in a way that the decrease in iNOS expression is a result of the decreased TNFα gene expression in CD26^−/−^ animals, whereas the increase in Arg-1 expression is an indirect result of an increased STAT6 expression that will be discussed later. In support of previously suggested activation of M2 macrophages under CD26 deficiency, an increase in Fizz gene expression has also been shown. As both Arg-1 and Fizz were shown to be affected by CD26 deficiency in a similar way, we could conclude that the absence of CD26 shifts the M1/M2 polarization towards the M2 anti-inflammatory phenotype.

During inflammation, many of the cytokines and growth factors present in the colon generate their cellular effects through one or more members of the STAT family of proteins by managing numerous physiologic cell processes, including macrophage polarization. As it was shown that the ablation of STAT6 annuls M2 gene expression and an enhanced expression of M2 genes in STAT6-overexpressing macrophages was noted, it is suggested that STAT6 mediates the transcriptional activation of M2 macrophage dependent genes bearing anti-inflammatory properties [35,36,37]. Our results further support the importance of STAT6-mediated signaling in M2 polarization, which we found to be more enhanced under CD26 deficiency. In addition to STAT6, the shown M2 phenotype can also be promoted through the activation of STAT3 [3], as supported by our results of increased STAT3 phosphorylation in CD26^−/−^ mice during colitis development. The opposite effect of CD26 deficiency was revealed in STAT1 gene expression that was found to be upregulated in C57BL/6 mice but, intriguingly, not in CD26^−/−^ mice. Contrary to that, STAT1 protein expression, as well as activation, did not differ between strains in acute colitis, indicating that CD26 deficiency is not a key factor in STAT1-mediated M1 activation.

To summarize, this study is among the first to highlight the role of CD26 in the process of macrophage polarization in UC. Our results indicate that CD26 deficiency directs the polarization of macrophages toward the M2 phenotype mediated by STAT6 and STAT3 activation, thus contributing to the anti-inflammatory response and disease reduction. As DP4 inhibitors are widely used for treating diabetes mellitus patients, further detailed investigations of the impact of CD26 inactivation during the activation of the immune response are of great importance.

## 4. Materials and Methods

### 4.1. Animals

Male, eight to ten-week-old mice with inactivated CD26 (CD26^−/−^), generated as previously described [38], and C57BL/6 were used in the study. C57BL/6 mice were purchased from the Central Animal Facility of the Faculty of Medicine, University of Rijeka, Croatia. All mice were housed in plastic cages and maintained under controlled environmental conditions: 12-h light/dark cycles, constant temperature of 20 ± 1 °C, and humidity of 50 ± 5%. They were fed with standard pellet food (MK, Complete Diet for Laboratory Rats and Mice, Ljubljana, Slovenia) and had free access to tap water. For the experiment, mice were randomly divided into experimental groups, each consisting of 6–8 mice.

### 4.2. Colitis Induction and Evaluation of Colitis Severity

Experimental colitis was induced by 3% (wt/vol) dextran sulfate sodium (DSS, 36–50 kDa, MP Biomedicals, Santa Ana, CA, USA) dissolved in drinking water for 7 days [39]. Treated mice had free access to the DSS solution, which was freshly prepared every other day. Control mice received tap water throughout the experimental period (Figure 1a). Results are representative of at least three independent experiments with a minimum of two l replicates per experiment. The general condition of the mice, changes in body weight, and development of clinical symptoms of colitis were observed and noted daily during DSS treatment, as well as during the recovery period that began with the termination of the DSS administration (day 7) and lasted until the end of the experiment (day 15). Stool samples were collected and evaluated for occult blood content using hemoccult blood test strips (Hemoccult II Dispensepak Plus Test, Beckman-Coulter, Brea, CA, USA).

The disease activity index (DAI) was calculated as previously described [19]. Briefly, DAI was calculated as the average of scores for weight loss, stool consistency, and fecal bleeding: DAI = (weight loss + stool consistency + fecal bleeding)/3. All mice were anesthetized by intraperitoneal administration of ketamine (2.5 mg/mouse) and sacrificed by cervical dislocation on days 3, 7, and 15. The colon was isolated by trimming at the ileocecal junction and the distal end of the rectum. The length of the colon was measured, and the mass was weighed. The colon was cut into small pieces that were fixed in 4% paraformaldehyde and embedded in paraffin or stored at −80 °C for further analysis. Additionally, blood samples were collected and processed as described below, and the resulting sera were stored.

### 4.3. Histological Evaluation and Immunohistochemical Staining

Paraffin-embedded colonic tissue samples were serially sectioned (4 µm in thickness) and stained with hematoxylin and eosin (HE) according to a standard protocol or processed for immunohistochemical analysis of Ki-67, DP8, and DP9 expression. Briefly, after deparaffinization and rehydration followed by heating in citrate buffer solution (0.01 M, pH 6.0), colonic tissue sections were incubated with 5% bovine serum albumin (BSA) for 1-h at room temperature. Sections were then incubated with primary antibody against Ki-67 (A5-14520, Invitrogen, Carlsbad, CA, USA) diluted in 3% BSA in phosphate-buffered saline (BSA/PBS) or with primary antibody against DP8 (MA5-25895, Invitrogen, Carlsbad, CA, USA) and DP9 (ab42080, Abcam, Cambridge, UK) diluted in 1% BSA/PBS in a humidified chamber at 4 °C overnight. The staining was performed by using Dako REAL™ EnVision™ Detection System, Peroxidase/DAB+, Rabbit/Mouse kit (K5007, Dako, Glostrup, Denmark) according to the manufacturer’s instructions. 3,3′-diaminobenzidine chromogen was used as the substrate yielding a brown-colored precipitate. Sections were then counterstained with hematoxylin, dehydrated, and cover-slipped with BioMount mounting media (Biognost, Zagreb, Croatia). Sections were visualized under a light microscope Olympus BX51 (Olympus Corporation, Tokyo, Japan) and images were captured and edited with Olympus DP70 digital camera (Olympus Corporation, Tokyo, Japan).

Histological analysis of colonic tissue was performed in a blinded fashion according to the previously published scoring criteria [40]: inflammation severity (none, 0; mild, 1; moderate, 2; severe, 3), the degree of inflammatory cell infiltration (normal, 0; mucosa, 1; mucosa plus submucosa, 2; transmural extension of the infiltrate, 3), the epithelial damage (intact, 0; crypt architecture distortion, 1; erosion, 2; ulceration, 3); the extent of lesions (none, 0; punctuate, 1; multifocal, 2; diffuse, 3), and edema score (none, 0; mild edema in the mucosa, 1; mucosa and submucosa, 2; the entire wall of the colon, 3). The microscopic damage score was calculated as the average of the sum of the total scores for the different parameters in each experimental group.

### 4.4. RNA Extraction and Quantitative Real-Time (RT)-PCR Analysis

Colonic tissue samples were homogenized with Polytron PT 1600E homogenizer (Kinematica AG, Luzern, Switzerland) and total RNA was extracted using TRI Reagent (Ambion, Austin, TX, USA) according to the manufacturer’s protocol. RNA was purified using the previously described lithium chloride precipitation method [41], quantified spectrophotometrically at 260 nm, and stored in diethylpyrocarbonate water (Ambion, Austin, TX, USA) at −80 °C. The integrity of isolated mRNA was analyzed by 2% agarose gel electrophoresis. A total of 2 µg of RNA was transcribed into cDNA using the High-capacity cDNA Reverse Transcription Kit (Applied Biosystems, Foster City, CA, USA). To exclude DNA contamination and unspecific amplification of gDNA, synthesis was additionally carried out with the same components, except for the reverse transcriptase. Quantification of gene expression was performed using the SYBR green Gene Expression Assay (Applied Biosystems, Foster City, CA, USA) and forward and reverse primers for each gene of interest which sequences are outlined in Table 1. Each reaction was performed in duplicate in a 25 µL reaction mixture containing 5 µg of cDNA with a 7300 RT-PCR System (Applied Biosystems). The relative changes in mRNA expression were calculated using the 2^−ΔΔCT^ method and normalized to the expression of β-actin or Rplp0 housekeeping genes.

### 4.5. Determination of MCP-1 Concentration

Colonic segments were homogenized in RIPA (RIPA Lysis buffer System, ChemCruz Biochemicals, Santa Cruz Biotechnology, Inc., Dallas, TX, USA; PhosStop EasyPack, Roche Diagnostics GmbH, Mannheim, Germany) containing protease and phosphatase inhibitors and the supernatant was collected after centrifugation at 12,000× *g* for 15 min at 4 °C. Blood samples were centrifuged at 15,000× *g* for 5 min at 18 °C and the separated serum was frozen at −80 °C until analysis. MCP-1 protein concentration in serum and colonic tissue was determined using an enzyme-linked immunosorbent assay (ELISA) kit (R&D Systems, Minneapolis, MN, USA) according to the manufacturer’s instructions. Each sample was run in triplicate. The concentration of MCP-1 in each sample was calculated from the standard curve and expressed in picograms per milliliter (pg/mL). The absorbance was measured by a microplate reader (Bio-Tek EL808, Winooski, VT, USA).

### 4.6. Isolation of Total Proteins and Western Blot Analysis

Colonic tissue segments were homogenized in ice-cold radioimmunoprecipitation assay (RIPA) lysis buffer with the addition of protease and phosphatase inhibitors. The supernatant containing total proteins was collected after centrifugation at 15,000× *g* for 20 min and the total protein concentration was determined using a bicinchoninic acid assay. An equal amount of proteins was separated by SDS-PAGE gel electrophoresis and transferred onto a polyvinylidene fluoride membrane. After blocking in 5% nonfat dry milk, the membranes were incubated overnight at 4 °C with primary antibodies against STAT1 (1:1000, #14994, Cell Signaling Technology, Danvers, MA, USA), phospho-STAT1 (Tyr701) (1:1000, #7649, Cell Signaling Technology, Danvers, MA, USA), STAT3 (1:1000, #12640, Cell Signaling Technology), phospho-STAT3 (Tyr705) (1:1000, ab76315, Abcam, Cambridge, UK), STAT6 (1:1000, #5397, Cell Signaling Technology, Danvers, MA, USA), phospho-STAT6 (Tyr641) (1:1000, #56554, Cell Signaling Technology, Danvers, MA, USA) and β-actin (1:10000, ab8226, Abcam, Cambridge, UK). The membranes were then incubated with appropriate secondary antibodies for 1-h at room temperature. Chemiluminescent signals were detected using SignalFire Elite ECL Reagent and scanned (Allianze 4.0, Cambridge, UK). Quantification of protein bands was performed using computer image analysis software ImageJ (U.S. NIH, Bethesda, Maryland, MD, USA).

### 4.7. Statistical Analysis

Statistical comparisons were made using STATISTICA version 12.0 (StatSoft Inc., Tulsa, OK, USA). Data were presented as mean ± standard deviation. Differences between groups have been tested using the one-way analysis of variance (ANOVA), followed by Scheffe’s post-hoc test. The level of *p* < 0.05 was considered significant.

## Figures and Tables

**Figure 1 ijms-23-05506-f001:**
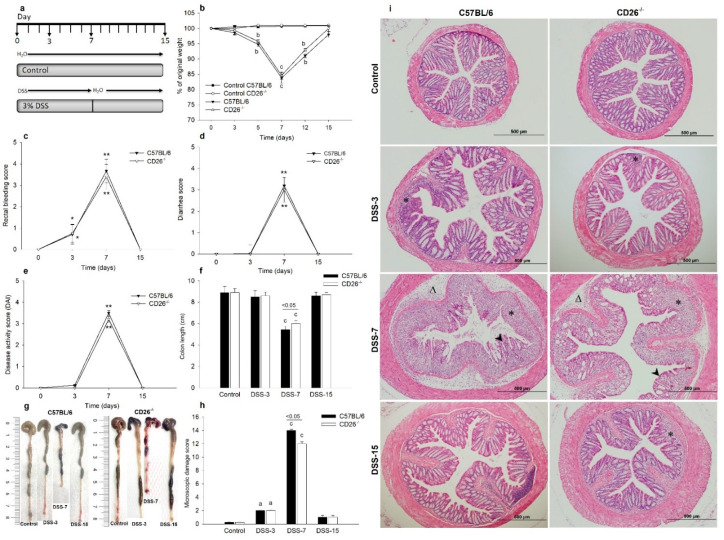
Clinical and histological changes during acute dextran sulfate sodium (DSS)-induced colitis and healing period in C57BL/6 and CD26-deficient (CD26^−/−^) mice. (**a**) Schematic illustration of the experimental protocol over time. C57BL/6 and CD26^−/−^ mice were administrated to DSS in drinking water from day 0 to day 7 followed by the administration of tap water during the recovery period until the end of the experiment on day 15. Mice were sacrificed on days 3, 7, and 15. The control group received tap water throughout the 15 days of the experiment. (**b**) Changes in the body weight were calculated as the percentage of initial body weight ± SD. (**c**) Rectal bleeding scores. (**d**) Diarrhea scores. (**e**) Disease activity scores (DAI). (**f**) The length of the colon. (**g**) Representative macroscopic images of the colon of DSS-treated C57BL/6 and CD26^−/−^ mice on days 3, 7, and 15 of the experiment and their respective controls. (**h**) Microscopic damage score of the colonic sections of control and DSS-treated C57BL/6 and CD26^−/−^ mice. (**i**) Representative images of hematoxylin and eosin staining of colonic tissue demonstrating microscopic changes and severity of inflammation during progression and resolution of colitis. Arrowhead indicates an area of distortion of crypts and mucosal damage; asterisk indicates infiltration of inflammatory cells and triangle indicates submucosal edema. Original magnification x100. Data are expressed as mean ± SD. *n* = 6 mice/each group. a, b and c denote *p* < 0.05, *p* < 0.01, and *p* < 0.001 compared with the control group, respectively. * *p* < 0.05, ** *p* < 0.01 comparing day 0 vs. day 3, 7, or 15. Scale bar: 500 μm.

**Figure 2 ijms-23-05506-f002:**
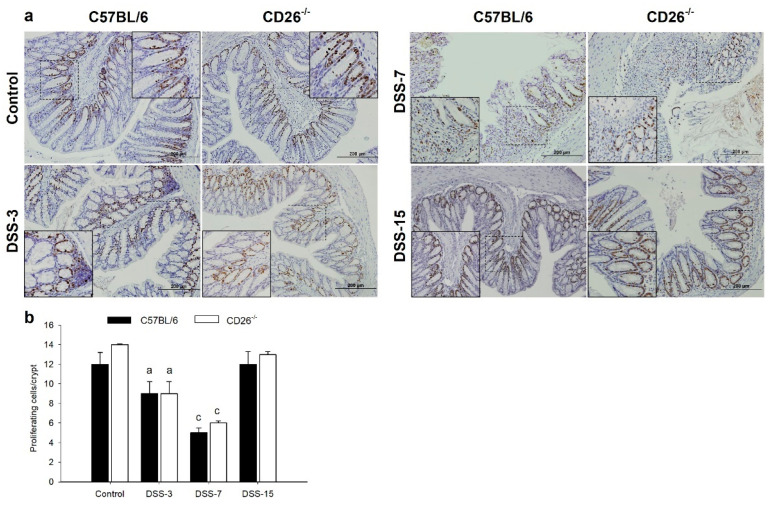
The influence of CD26 deficiency on the proliferative activity of the colonic cells during progression and resolution of dextran sulfate sodium (DSS)-induced colitis in C57BL/6 and CD26-deficient (CD26^−/−^) mice. (**a**) Representative images of immunohistochemical analysis of the expression of Ki-67 cell proliferation marker in colonic tissue of C57BL/6 and CD26^−/−^ mice treated with DSS. (**b**) Quantitative analysis of Ki-67 positive cells within the colonic crypts. Original magnification ×200. Insets represent digital zooms of the region defined by the dotted line. Scale bar: 200 μm. Data are expressed as mean ± SD. *n* = 6 mice/each group. a and c denote *p* < 0.05 and *p* < 0.001 compared with the control group, respectively.

**Figure 3 ijms-23-05506-f003:**
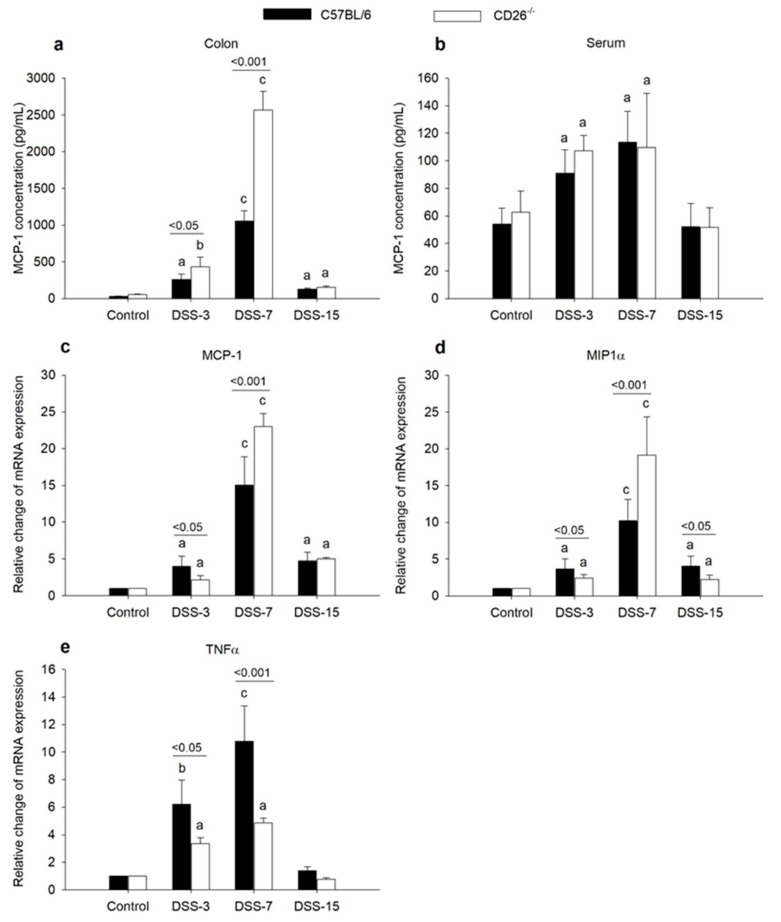
The influence of CD26 deficiency on concentration and mRNA expression of proinflammatory mediators MCP-1 concentration in colonic tissue (**a**) and serum (**b**) of C57BL/6 and CD26^−/−^ mice with DSS-induced colitis was determined using enzyme-linked immunosorbent assay (ELISA). The analysis was performed on days 3, 7, and 15 after colitis induction with DSS in both mouse strains. Relative quantification of MCP-1 (**c**), MIP1α (**d**), and TNFα (**e**) gene expression in colonic tissue was determined by quantitative real-time (RT)-PCR on days 3, 7, and 15 after colitis induction. Data are expressed as mean ± SD. *n* = 6 mice/each group. a, b, and c denote *p* < 0.05, *p* < 0.01 and *p* < 0.001 compared with the control group, respectively. Comparisons between related DSS-treated C57BL/6 and CD26^−/−^ groups are highlighted by horizontal lines above the bars.

**Figure 4 ijms-23-05506-f004:**
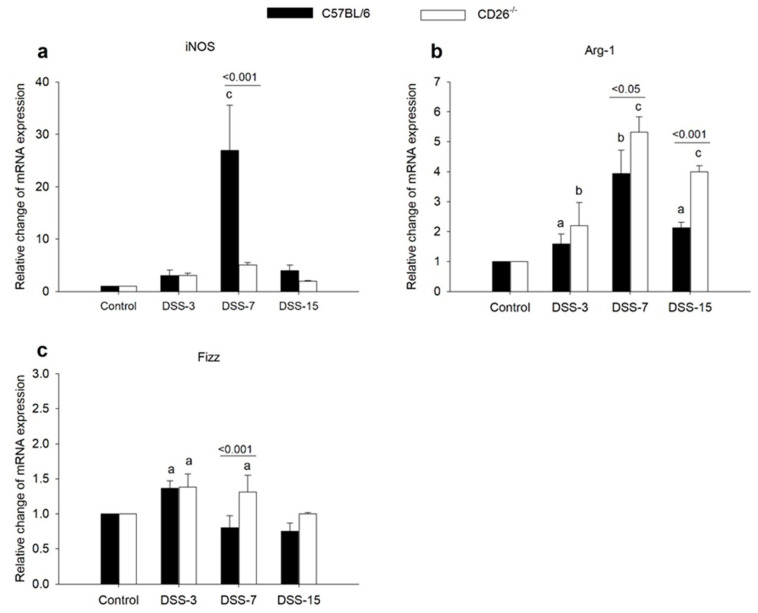
The influence of CD26 deficiency on mRNA expression of macrophage markers in the colon of DSS-treated mice. Relative quantification of iNOS (**a**), Arg-1 (**b**), and Fizz (**c**) genes expression in colonic tissue of C57BL/6 and CD26^−/−^ mice with DSS-induced colitis was determined by quantitative real-time (RT)-PCR on days 3, 7, and 15 days after colitis induction. The RT-PCR analysis was performed in duplicate. Data are expressed as mean ± SD. *n* = 6 mice/each group. a, b, and c denote *p* < 0.05, *p* < 0.01, and *p* < 0.001 compared with the control group, respectively. Comparisons between related DSS-treated C57BL/6 and CD26^−/−^ groups are highlighted by horizontal lines above the bars.

**Figure 5 ijms-23-05506-f005:**
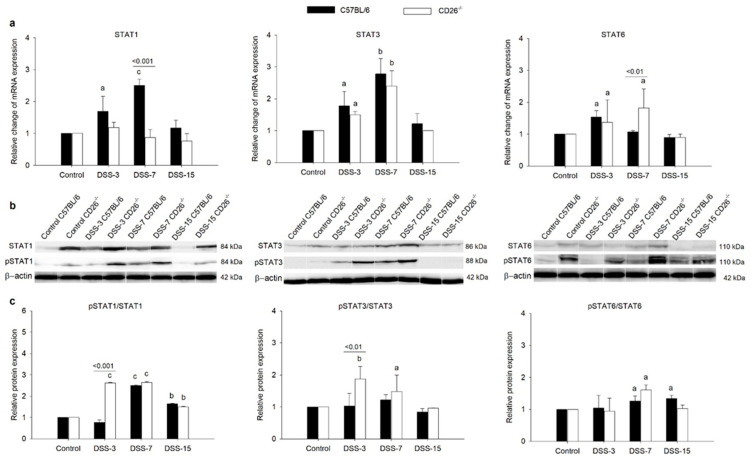
The influence of CD26 deficiency on mRNA and protein expression of STAT transcription factors in colonic tissue during dextran sulfate sodium (DSS)-induced colitis (**a**) Relative quantification of STAT1, STAT3, and STAT6 genes expression in colonic tissue of C57BL/6 and CD26^−/−^ mice with DSS-induced colitis was determined by quantitative real-time (RT)-PCR on days 3, 7, and 15 days after colitis induction. The RT-PCR analysis was performed in duplicate. (**b**) Protein expression of STAT1, STAT3, and STAT6 and their phosphorylated forms in colonic tissue of C57BL/6 and CD26^−/−^ mice with DSS-induced colitis was performed by Western blot analysis on days 3, 7, and 15 days after colitis induction. (**c**) Relative quantification of the expression ratio of proteins phospho-STAT1 and STAT1, phospho-STAT3 and STAT3, and phospho-STAT6 and STAT6. Data are expressed as mean ± SD. *n* = 6 mice/each group. a, b, and c denote *p* < 0.05, *p* < 0.01, and *p* < 0.001 compared with the control group, respectively. Comparisons between related DSS-treated C57BL/6 and CD26^−/−^ groups are highlighted by horizontal lines above the bars.

**Figure 6 ijms-23-05506-f006:**
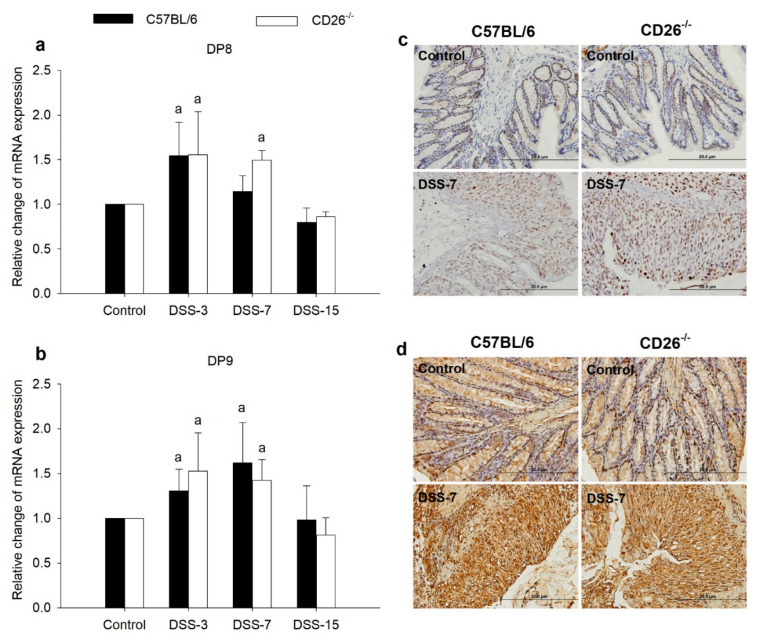
Expression of DP8 and DP9 in colonic tissue of CD26-deficient (CD26^−/−^) mice with dextran sulfate sodium (DSS)-induced colitis. Relative mRNA expression of DP8 (**a**) and DP9 (**b**) in colonic tissue of C57BL/6 and CD26^−/−^ mice treated with DSS was assessed by quantitative real-time (RT)-PCR. Representative images of immunohistochemical analysis of DP8 (**c**) and DP9 (**d**) protein expressions in colonic tissue of DSS-treated C57BL/6 and CD26^−/−^ mice in the acute phase of colitis (day 7). Original magnification x400. Scale bar: 20 μm. The RT-PCR assay was performed in duplicate. Data are presented as mean ± SD. *n* = 6 mice/each group. a: *p* < 0.05 compared with control group.

**Table 1 ijms-23-05506-t001:** Forward and reverse primer sequences used in quantitative real-time (RT)-PCR analysis of the expression of genes of interest.

Gene	Forward Primer	Reverse Primer
MCP1	5′TTAAAAACCTGGATCGGAACCAA3′	5′GCATTAGCTTCAGATTTACGGGT3′
MIP-1α	5′CATATGGAGCTGACACCCCG3′	5′GTCAGGAAAATGACACCTGGC3′
TNF-α	5′CCTGTAGCCCACGTCGTAG3′	5′GGGAGTAGACAAGGTACAACCC3′
iNOS	5′AGTCAACTGCAAGAGAACGGA3′	5′GGCTGAGAACAGCACAAGGG3′
Arg	5′TGGCTTGCGAGACGTAGAC3′	5′GCTCAGGTGAATCGGCCTTTT3′
Fizz	5′TCCCTCCACTGTAACGAAGAC3′	5′CTCCCAAGATCCACAGGCAA3′
STAT1	5′TCACAGTGGTTCGACCTTCAG3′	5′GCAAACGAGACATCATAGGCA3′
STAT3	5′AGAACCTCCAGGACGACTTTG3′	5′TCACAATGCTTCTCCGCACTCT3′
STAT6	5′GACCTGTCCATTCGCTCACT3′	5′GGATGACGTGTGCAATGGTG3′
DP8	5′CCCAAGCGGAAAGAACTCCT3′	5′CCAACATGGGGGACGTAACA3′
DP9	5′CTGGATCAACGTCCACGACA3′	5′AGGGGTTCCGTCCAGTCATA3′
β-actin	5′GGCTGTATTCCCCTCCATCG3′	5′CCAGTTGGTAACAATGCCATGT3′
Rplp0	5′AGATTCGGGATATGCTGTTGGC3′	5′TCGGGTCCTAGACCAGTGTTC3′

## Data Availability

The data that support the findings of this study are available from the corresponding author upon reasonable request.

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
