# Peer review of "CD26 Deficiency Controls Macrophage Polarization Markers and Signal Transducers during Colitis Development and Resolution"

_ijms, 2022, doi:10.3390/ijms23105506_

Round 1

Reviewer 1 Report

 The study in  CD26-/- mice treated with 3% DSS solution to induce colitis reported by the authors describe cleary the effect of  CD26 deficiency to directs polarization of macrophages to the M2 phenotype mediated by STAT6 and STAT3 activation thus contributing to the anti-inflammatory  responce. The paper is well written and fluid and section methods are well detailed. I suggest minor modification in the figure for example the authors could  add molecular weight to blots and provide enlarged cropped images. I strongly encourage to extend the study on human experimental models

Author Response

The response to reviewer’s comments and suggestions for authors:

Reviewer #1:

Comment: The study in CD26-/- mice treated with 3% DSS solution to induce colitis reported by the authors describe clearly the effect of CD26 deficiency to directs polarization of macrophages to the M2 phenotype mediated by STAT6 and STAT3 activation thus contributing to the anti-inflammatory response. The paper is well written, and fluid and section methods are well detailed. I suggest minor modification in the figure for example the authors could add molecular weight to blots and provide enlarged cropped images. I strongly encourage to extend the study on human experimental models.

Thank you for your suggestion regarding the improvement of Western blot figures, we have added the molecular weight of each molecule to Figure 5 and enlarged blot images. Furthermore, we plan to expand our study on the patients suffering from inflammatory bowel disease to additionally test our findings of the impact of CD26 inhibition on macrophage polarization in ulcerative colitis.

Reviewer 2 Report

In this manuscript Vukelic and colleagues examined the role of CD26 deficiency in a murine DSS induced colitis model.

  • In general: it is not clear how often the experiments where performed. Please clearly Indicate how often the experiments were performed and if “wt” mice are littermates or C57BL/6 as it is stated in the Material and Method part. If no littermates were used, “wt” should be changed into C57BL/6 in the manuscript
  • Statistics: why did the authors not distinguish between * = p ≤05, ** = p ≤ 0.01;  *** = p ≤ 0.001? They just mentioned in the statistic explanation “the level of p < 0.05 was considered as significant”. For me it is not clear if no data did reach the ** = p ≤ 0.01 or more or if they did not make difference in the significance? That would be relevant to interpret the data in the right way.
  • Page 2 line 54-55, the authors state “the regulation of M1/M2 balance has recently been targeted as potential therapeutic strategy for UC”. Please provide a reference
  • Page 3 line 104-106: The authors state that „ Additionally, our goal was to assess changes in the expression pattern of DP8 and DP9 in the colonic tissue during the development and resolution of experimental colitis under CD26 deficiency”. The same working group already examined this in detail, even though a TNBS induced colitis model and not a DSS colitis model was used (Reference 19, Buljevic et al. 2017 Journal of cellular Biochemistry). So in my opinion the novelty is lacking.
  • Figure 1: The authors already publish more or less the same setting (Reference 20, Detel et al. 2012 Exp Physiol), so I do not see any novelty here. Most of the (slight) differences are between untreated (no DSS) and respective DSS treated groups and not between wt and CD26-/- DSS treated groups. So it is not surprising that the colon length for example is shorter in DSS treated mice compared to untreated control groups
  • Figure 1c – Figure 1e: The control groups (no DSS) are not depicted, but the statistics was performed with “a” = compared with control group
  • Figure 1i: It might be due to the pdf resolution, but the representative histological pictures are not very sharp, so it is hard to follow the description in the text
  • Figure 2: Here the authors only show, that in the course of DSS, the proliferating cells/crypt decrease in wt and CD26-/- compared to the untreated (no DSS) groups. There are no changes between the wt and CD26-/-. So please clearly that this in the manuscript. A similar comparison between untreated and DSS treated mice was for example already shown by Araki et al 2021, Oncol Rep, PubMed ID: 22895560)
  • Figure 3: it would be interesting to the how other macrophage / IBD related pro-inflammatory cytokines like IL1ß or IL6 would behave in this setting. Did the authors also check for macrophages via flow cytometry? Here they could easily analyze the distribution/ratio between “M1 and M2” macrophages with several markers
  • The authors could also isolated CD14+ monocytes from wt and CD26-/- for in vitro polarization in vitro (into M1/M2) and analyze if there are any differences in polarization and cytokine expression.
  • Figure 4a: The bars for standard deviation for CD26-/- on day 7 and 15 seem to be missing
  • Figure 5A: The results of the RT-PCR should be interpreted with caution. For most significant differences, the fold change is not more than 2. Maybe this should be mentioned in the text / heading
  • Figure 5C: If the values are relative, why are there some bars with standard deviation and others seem not to have a standard deviation in the control group? Please explain
  • Figure 6: As the authors mentioned correctly there is no difference for DP8 and DP9 between wt and CD26-/- in DSS induced colitis only an increase compared to the control. The authors have already checked for the role of DP8 and DP9 in a TNBS colitis model (Reference 19, Buljevic et al. 2017 Journal of cellular Biochemistry). Here, the tendency was, that DP8 and DP) are decreased compared to untreated control. Please explain. Again in my opinion, a real novelty is lacking and it is just a repetition of already performed experiments in another colitis model. It would be more valuable to see effects in a chronic setting of colitis.
  • Discussion: page 10 line 290 – 291: Again the effects are very slight and not for all parameters and only for single time points. So please clearly indicate this in the discussion
  • Discussion: page 10 line 300 – 302: I do not agree with this sentence. Here it should be mentioned that CD26-/- was increased in comparison to untreated group! Please clearly indicate for DP8 that there are no significant differences between DSS treated WT and CD26-/- , similar as for DP9
  • Discussion: page 11 line 386 summary. The summary is overstated and should be rephrased: The authors did not perform any experiments with UC patients or material

Author Response

The response to the reviewer’s comments and suggestions for authors:

Reviewer #2:

In this manuscript, Vukelic and colleagues examined the role of CD26 deficiency in a murine DSS induced colitis model.

In general: it is not clear how often the experiments where performed. Please clearly Indicate how often the experiments were performed and if “wt” mice are littermates or C57BL/6 as it is stated in the Material and Method part. If no littermates were used, “wt” should be changed into C57BL/6 in the manuscript.

As is written in the Material and Methods section (4.2. Colitis induction and evaluation of colitis severity), experiment was designed to expose the animals to the DSS solution for 7 days and after that period mice drank regular water. In that maximum of 15 days, mice were sacrificed on days 3, 7 or 15 which represented different time points during colitis development. Experiments began in a point when male mice of the same litter reached the age of 8 - 10 weeks and were conducted until all our experimental groups comprised of 6 – 8 animals.

The term “WT” refers to wild-type mice of C57BL/6 origin and, according to the reviewer’s suggestion, we have changed “WT” to “C57BL/6” throughout the manuscript as well as in all the Figures.

Statistics: why did the authors not distinguish between * = p ≤05, ** = p ≤ 0.01;  *** = p ≤ 0.001? They just mentioned in the statistic explanation “the level of p < 0.05 was considered as significant”. For me it is not clear if no data did reach the ** = p ≤ 0.01 or more or if they did not make difference in the significance? That would be relevant to interpret the data in the right way.

Thank you for your comment. The level of statistical significance was set to p < 0.05. The majority of the results reached that value and were therefore considered to be relevant.

Page 2 line 54-55, the authors state “the regulation of M1/M2 balance has recently been targeted as potential therapeutic strategy for UC”. Please provide a reference

In line with your suggestion the reference for the statement „The regulation of M1/M2 balance has recently been targeted as potential therapeutic strategy for UC“ was added:

Long J, Liu XK, Kang ZP, Wang MX, Zhao HM, Huang JQ, Xiao QP, Liu DY, Zhong YB. Ginsenoside Rg1 ameliorated experimental colitis by regulating the balance of M1/M2 macrophage polarization and the homeostasis of intestinal flora. Eur J Pharmacol. 2022 Feb 15;917:174742. doi: 10.1016/j.ejphar.2022.174742. Epub 2022 Jan 6. PMID: 34999087.

Page 3 line 104-106: The authors state that „ Additionally, our goal was to assess changes in the expression pattern of DP8 and DP9 in the colonic tissue during the development and resolution of experimental colitis under CD26 deficiency”. The same working group already examined this in detail, even though a TNBS induced colitis model and not a DSS colitis model was used (Reference 19, Buljevic et al. 2017 Journal of cellular Biochemistry). So in my opinion the novelty is lacking.

We thank the reviewer for this suggestion. It is true that we have determined DP8 and DP9 expression in an experimental model called TNBS colitis. However, that model mimics Crohn’s disease, the other type of inflammatory bowel disease (IBD) that has some similarities with ulcerative colitis but also some very distinguished differences. As we found that TNBS-colitis induction causes changes in the expression profiles of the two proteins belonging to the CD26 family, we wanted to determine whether the same changes are present also in the DSS-induced colitis to get a wider picture on their role in IBD.

Figure 1: The authors already publish more or less the same setting (Reference 20, Detel et al. 2012 Exp Physiol), so I do not see any novelty here. Most of the (slight) differences are between untreated (no DSS) and respective DSS treated groups and not between wt and CD26-/- DSS treated groups. So it is not surprising that the colon length for example is shorter in DSS treated mice compared to untreated control groups

The purpose of Figure 1 was to determine a successful establishment of colitis based on the clinical, macroscopic, and histological assessment as well as to depict a schematic illustration of the experimental protocol. Although the results regarding body mass variations, DAI, and histopathological changes presented in Figure 1 are somewhat expected, the authors feel (and some reviewers strictly recommend) that they represent an unquestionably necessary part of the Results section and provide proof that the model has been efficiently set up.

Figure 1c – Figure 1e: The control groups (no DSS) are not depicted, but the statistics was performed with “a” = compared with control group

In Figures 1c (Rectal bleeding scores), 1d (Diarrhea scores), and 1e (Disease activity scores), the data regarding the control group are not depicted because control group mice do not have any of the aforementioned symptoms as they are healthy. To better clarify the issue, a sentence was added on page 4, line 138: “Control mice which received tap water did not develop previously mentioned clinical symptoms”.

Figure 1i: It might be due to the pdf resolution, but the representative histological pictures are not very sharp, so it is hard to follow the description in the text.

The representative histological pictures were improved as much as it was possible. Since images were converted to pdf, a part of the image sharpness has been lost but we believe that the online version will be sharp.

Figure 2: Here the authors only show, that in the course of DSS, the proliferating cells/crypt decrease in wt and CD26-/- compared to the untreated (no DSS) groups. There are no changes between the wt and CD26-/-. So please clearly that this in the manuscript. A similar comparison between untreated and DSS treated mice was for example already shown by Araki et al 2021, Oncol Rep, PubMed ID: 22895560)

The suggestion was accepted completely and included in the manuscript (page 4, line 157).

Figure 3: it would be interesting to the how other macrophage / IBD related pro-inflammatory cytokines like IL1ß or IL6 would behave in this setting. Did the authors also check for macrophages via flow cytometry? Here they could easily analyze the distribution/ratio between “M1 and M2” macrophages with several markers

The authors thank for this valuable suggestion, we will include flow cytometry in future research where we plan to expand our findings.

The authors could also isolated CD14+ monocytes from wt and CD26-/- for in vitro polarization in vitro (into M1/M2) and analyze if there are any differences in polarization and cytokine expression.

The authors thank for this valuable suggestion.

Figure 4a: The bars for standard deviation for CD26-/- on day 7 and 15 seem to be missing

The bars for CD26-/- on days 7 and 15 were added. However, in some cases the values for standard deviations for both mouse strains are very low therefore the bars are barely noticeable.

Figure 5A: The results of the RT-PCR should be interpreted with caution. For most significant differences, the fold change is not more than 2. Maybe this should be mentioned in the text / heading

The Figure 5a shows protein expressions of STAT1, STAT36, and STAT6. The most interesting changes (and differences) were noted on day 7 which is our key experimental time point with values of fold change being closer to 3 than to 2 and those differences for discussed the most.

Figure 5C: If the values are relative, why are there some bars with standard deviation and others seem not to have a standard deviation in the control group? Please explain

We thank the reviewer for this suggestion, the error was corrected and the lines representing standard deviation that were added to some bars by mistake were removed.

Figure 6: As the authors mentioned correctly there is no difference for DP8 and DP9 between wt and CD26-/- in DSS induced colitis only an increase compared to the control. The authors have already checked for the role of DP8 and DP9 in a TNBS colitis model (Reference 19, Buljevic et al. 2017 Journal of cellular Biochemistry). Here, the tendency was, that DP8 and DP) are decreased compared to untreated control. Please explain. Again in my opinion, a real novelty is lacking and it is just a repetition of already performed experiments in another colitis model. It would be more valuable to see effects in a chronic setting of colitis.

In our paper mentioned by the reviewer, we have determined the effect of CD26 deficiency on DP8 and DP9 expression to investigate whether in the absence of CD26 one or both enzymes take on its enzymatic and/or co-stimulatory roles. We have found a significant change in DP9 expression in acute colitis in C57BL/6 mice, suggesting exactly our hypothesis. As DSS-colitis mimics ulcerative colitis rather than Crohn’s disease, our goal was to find out if that effect is also present in that model due to differences in the nature of the activated immune response and we concluded that it is not. We will take into consideration the reviewer’s suggestion for testing DP8 and DP9 in a chronic colitis model.

Discussion: page 10 line 290 – 291: Again the effects are very slight and not for all parameters and only for single time points. So please clearly indicate this in the discussion

According to the suggestion, it was emphasized and clearly indicated in the discussion that effects were not noted for all parameters and only for some of the time points.

Therefore, the following sentence was added at page 11, line 305-307: “However, it should be emphasized that the observed effects were relatively slight, for certain time points and were not noted for all parameters at same level of significance”.

Discussion: page 10 line 300 – 302: I do not agree with this sentence. Here it should be mentioned that CD26-/- was increased in comparison to untreated group! Please clearly indicate for DP8 that there are no significant differences between DSS treated WT and CD26-/- , similar as for DP9

In line with your suggestion the change was made accordingly (page 11, line 316-320).

Discussion: page 11 line 386 summary. The summary is overstated and should be rephrased: The authors did not perform any experiments with UC patients or materia.

We have taken into consideration the reviewer’s comment and rephrased parts of the final paragraph of the Discussion section. We would also like to mention that the other reviewers asked to emphasize the possible role of CD26 inhibition in the treatments for UC.

Reviewer 3 Report

This manuscript presents interesting data regarding the pathogenetic mechanism of UC and the possible role of CD 26 deficiency and macrophage polarization. 

The paper is well written, with significant results that support the conclusions. The use of English is correct; there are no errors or editing mistakes. The study is easy to be followed. 

I have just one comment regarding the clinical importance of the author's findings. It would be better to add more data to the "Discussion" about the possible role in the new treatments for UC based on the findings. The author mentioned the role of DP4 inhibitors in their conclusions, but this idea may be developed more in Discussions. 

Author Response

The response to the reviewer’s comments and suggestions for authors:

Reviewer #3:

This manuscript presents interesting data regarding the pathogenetic mechanism of UC and the possible role of CD 26 deficiency and macrophage polarization.

The paper is well written, with significant results that support the conclusions. The use of English is correct; there are no errors or editing mistakes. The study is easy to be followed.

I have just one comment regarding the clinical importance of the author's findings. It would be better to add more data to the "Discussion" about the possible role in the new treatments for UC based on the findings. The author mentioned the role of DP4 inhibitors in their conclusions, but this idea may be developed more in Discussions.

The authors thank the reviewer for a very valuable and constructive suggestion. However, we feel that our present work represents only the beginning in the research of the effect that CD26 inhibition bears on the immunoinflammatory response developing in ulcerative colitis. As we did not include patients in this research, we feel that it would be too presumptuous to further discuss such ideas in this point, but we plan to do so in more detail when we test our hypothesis on human samples.

Round 2

Reviewer 2 Report

Some of the reviewers questions were answered properly but there are still some open questions which needs to be addressed by the authors. For a better overview I deleted all questions that were answered properly and added the comments in red and underlined to the open questions.

In general: it is not clear how often the experiments where performed. Please clearly Indicate how often the experiments were performed and if “wt” mice are littermates or C57BL/6 as it is stated in the Material and Method part. If no littermates were used, “wt” should be changed into C57BL/6 in the manuscript.

As is written in the Material and Methods section (4.2. Colitis induction and evaluation of colitis severity), experiment was designed to expose the animals to the DSS solution for 7 days and after that period mice drank regular water. In that maximum of 15 days, mice were sacrificed on days 3, 7 or 15 which represented different time points during colitis development. Experiments began in a point when male mice of the same litter reached the age of 8 - 10 weeks and were conducted until all our experimental groups comprised of 6 – 8 animals.

The term “WT” refers to wild-type mice of C57BL/6 origin and, according to the reviewer’s suggestion, we have changed “WT” to “C57BL/6” throughout the manuscript as well as in all the Figures.

*For me this question is not answered properly and just a description of the experimental set up itself. It is not clear if the results were confirmed in INDEPENDENT experiments. Please clarify    

Statistics: why did the authors not distinguish between * = p ≤05, ** = p ≤ 0.01;  *** = p ≤ 0.001? They just mentioned in the statistic explanation “the level of p < 0.05 was considered as significant”. For me it is not clear if no data did reach the ** = p ≤ 0.01 or more or if they did not make difference in the significance? That would be relevant to interpret the data in the right way.

Thank you for your comment. The level of statistical significance was set to p < 0.05. The majority of the results reached that value and were therefore considered to be relevant.

*The question is not answered properly in my opinion. Are there any data in the manuscript that are higher significant than * = p ≤0.05?  If so, this should be visible for a better interpretation of the data.

Page 3 line 104-106: The authors state that „ Additionally, our goal was to assess changes in the expression pattern of DP8 and DP9 in the colonic tissue during the development and resolution of experimental colitis under CD26 deficiency”. The same working group already examined this in detail, even though a TNBS induced colitis model and not a DSS colitis model was used (Reference 19, Buljevic et al. 2017 Journal of cellular Biochemistry). So in my opinion the novelty is lacking.

We thank the reviewer for this suggestion. It is true that we have determined DP8 and DP9 expression in an experimental model called TNBS colitis. However, that model mimics Crohn’s disease, the other type of inflammatory bowel disease (IBD) that has some similarities with ulcerative colitis but also some very distinguished differences. As we found that TNBS-colitis induction causes changes in the expression profiles of the two proteins belonging to the CD26 family, we wanted to determine whether the same changes are present also in the DSS-induced colitis to get a wider picture on their role in IBD.

*In many manuscripts, where only murine data are shown, it is mandatory to show 2 different murine colitis models. Even though some further aspect are explored in this DSS setting compared to the former published TNBS setting,  I do think that just repeating a very similar setting with another model provides significant novelty.  As also proposed, the authors should for example add data on a chronic model of DSS in this manuscript.

Figure 1c – Figure 1e: The control groups (no DSS) are not depicted, but the statistics was performed with “a” = compared with control group

In Figures 1c (Rectal bleeding scores), 1d (Diarrhea scores), and 1e (Disease activity scores), the data regarding the control group are not depicted because control group mice do not have any of the aforementioned symptoms as they are healthy. To better clarify the issue, a sentence was added on page 4, line 138: “Control mice which received tap water did not develop previously mentioned clinical symptoms”.

Yes, but then the statistics needs to be addressed with “b” and not with “a”, as the comparison is done between the DSS treated groups and not the untreated healthy group

Author Response

The response to reviewer’s comments:

Some of the reviewers questions were answered properly but there are still some open questions which needs to be addressed by the authors. For a better overview I deleted all questions that were answered properly and added the comments in red and underlined to the open questions.

1st revision

In general: it is not clear how often the experiments where performed. Please clearly Indicate how often the experiments were performed and if “wt” mice are littermates or C57BL/6 as it is stated in the Material and Method part. If no littermates were used, “wt” should be changed into C57BL/6 in the manuscript.

As is written in the Material and Methods section (4.2. Colitis induction and evaluation of colitis severity), experiment was designed to expose the animals to the DSS solution for 7 days and after that period mice drank regular water. In that maximum of 15 days, mice were sacrificed on days 3, 7 or 15 which represented different time points during colitis development. Experiments began in a point when male mice of the same litter reached the age of 8 - 10 weeks and were conducted until all our experimental groups comprised of 6 – 8 animals.

The term “WT” refers to wild-type mice of C57BL/6 origin and, according to the reviewer’s suggestion, we have changed “WT” to “C57BL/6” throughout the manuscript as well as in all the Figures.

2nd revision

*For me this question is not answered properly and just a description of the experimental set up itself. It is not clear if the results were confirmed in INDEPENDENT experiments. Please clarify    

 *Thank you for your comment. We apologize for not previously understanding the comment properly. According to the suggestion, it was emphasized and indicated how many independent experiments were done. Therefore, the sentence was added to the Materials and Methods, subsection 4.2. Colitis induction and evaluation of colitis severity (page 13).

1st revision

Statistics: why did the authors not distinguish between * = p ≤05, ** = p ≤ 0.01;  *** = p ≤ 0.001? They just mentioned in the statistic explanation “the level of p < 0.05 was considered as significant”. For me it is not clear if no data did reach the ** = p ≤ 0.01 or more or if they did not make difference in the significance? That would be relevant to interpret the data in the right way.

Thank you for your comment. The level of statistical significance was set to p < 0.05. The majority of the results reached that value and were therefore considered to be relevant.

2nd revision

*The question is not answered properly in my opinion. Are there any data in the manuscript that are higher significant than * = p ≤0.05?  If so, this should be visible for a better interpretation of the data.

*The request was accepted completely and statistics was redone according to your suggestion. To improve the interpretation of the data, all figures and figure legends have been changed suitably: a clear distinction between p≤0.05, p≤0.01, and p≤0.001 was highlighted in the figures and a detailed explanation was added to figure legends.

1st revision

Page 3 line 104-106: The authors state that „ Additionally, our goal was to assess changes in the expression pattern of DP8 and DP9 in the colonic tissue during the development and resolution of experimental colitis under CD26 deficiency”. The same working group already examined this in detail, even though a TNBS induced colitis model and not a DSS colitis model was used (Reference 19, Buljevic et al. 2017 Journal of cellular Biochemistry). So in my opinion the novelty is lacking.

We thank the reviewer for this suggestion. It is true that we have determined DP8 and DP9 expression in an experimental model called TNBS colitis. However, that model mimics Crohn’s disease, the other type of inflammatory bowel disease (IBD) that has some similarities with ulcerative colitis but also some very distinguished differences. As we found that TNBS-colitis induction causes changes in the expression profiles of the two proteins belonging to the CD26 family, we wanted to determine whether the same changes are present also in the DSS-induced colitis to get a wider picture on their role in IBD.

2nd revision

*In many manuscripts, where only murine data are shown, it is mandatory to show 2 different murine colitis models. Even though some further aspect are explored in this DSS setting compared to the former published TNBS setting,  I do think that just repeating a very similar setting with another model provides significant novelty.  As also proposed, the authors should for example add data on a chronic model of DSS in this manuscript.

*The aim of the study was the investigate changes of selected chemoattractants, macrophage polarization markers, transcriptional factors, and DP8/9 expression during the development and resolution of acute ulcerative colitis. In accordance, unfortunately, we can not improve the current manuscript with data regarding the expression of DP8 and 9, or any other investigated molecule, during chronic colitis development and resolution, although we hope to do so in near future.

Figure 1c – Figure 1e: The control groups (no DSS) are not depicted, but the statistics was performed with “a” = compared with control group

In Figures 1c (Rectal bleeding scores), 1d (Diarrhea scores), and 1e (Disease activity scores), the data regarding the control group are not depicted because control group mice do not have any of the aforementioned symptoms as they are healthy. To better clarify the issue, a sentence was added on page 4, line 138: “Control mice which received tap water did not develop previously mentioned clinical symptoms”.

2nd revision

Yes, but then the statistics needs to be addressed with “b” and not with “a”, as the comparison is done between the DSS treated groups and not the untreated healthy group

*The request is completely accepted. In line with the suggestion, the indication of the statistical significance between groups in Figures 1c to 1e has been changed. It is pointed out that the comparison was done between the non-treated (day 0) and the DSS treated groups (days 3, 7, and 15). The figure legend was changed accordingly (page 4). 

Round 3

Reviewer 2 Report

The authors have answered all questions properly but should perform additionally for the future experiments with more variability and not just "repeating" very similar (already published) settings.